# ANODIT: MASK GUIDED DIT INPAINT MODELS FOR ANOMALY IMAGES GENERATION

## ABSTRACT

Effective training of industrial anomaly detection (AD) models is persistently hindered by the scarcity and limited diversity of real anomaly samples. While generative methods have been proposed to augment anomaly data, they often struggle with a critical trade-off between generation controllability, background fidelity, and the realism of the synthesized anomalies. In this paper, we propose AnoDiT, a novel mask-guided anomaly generation framework that leverages a Diffusion Transformer (DiT) for high-fidelity inpainting. To ensure the perceptual plausibility of generated anomalies, we introduce a Laplacian Pyramid-based Texture Decomposition Module. which guides the model to learn deep texture representations of anomalous regions. Furthermore, for seamless integration of the anomaly into the pristine background, we design an Anomaly Region Focusing mechanism with Edge Weighting, which encourages the model to learn a natural transition at the defect boundary and is enhanced by a multi-round resampling process. To establish a fully automated pipeline and overcome the annotation bottleneck, we also develop a conditional diffusion model incorporating a Positional Prior to generate diverse and realistically-located anomaly masks. This dual-model pipeline not only enables fine-grained control over the anomaly's geometry and texture but also simultaneously yields pixel-perfect labels. Experiments demonstrate that data synthesized by AnoDiT significantly improves the performance of downstream anomaly inspection tasks.

## 1 INTRODUCTION

Data scarcity poses a significant impediment to the development of deep learning models for industrial AD. In contrast to widely available public datasets, data in industrial applications is characterized by a distinct long-tail distribution (Zhang et al., 2024). The scarcity of industrial anomaly data is mainly due to the rarity and unpredictability of real faults in production systems. In response, most AD methodologies are predicated on unsupervised learning (You et al., 2022; Lu et al., 2023) using normal data or employ few-shot learning strategies (Fang et al., 2023; Wang et al., 2022; Li et al., 2024b). Nevertheless, the deficiency of abnormal samples for training remains a persistent challenge.

Recent empirical and theoretical work has argued that Synthesizing anomalies with generative models is a promising alternative (Savage, 2023). In Fig.1, synthesizing anomalies have progressed from simple image transformations(Lin et al., 2021) to generative models based on Generative Adversarial Networks (GAN) (Niu et al., 2020; 2022b; Duan et al., 2023) and diffusion models (Jin et al., 2025), and vision-language models (Hu et al., 2024; Dai et al., 2024) guided by text prompts. However, approaches that decouple the anomaly from its context often degrade background quality by introducing unrealistic blending or artifacts. Conversely, holistic image generation is complex, lacks fine-grained control, and makes it difficult to balance the learning of background and foreground features. Furthermore, existing techniques offer limited control over crucial anomaly attributes like texture, shape, and location. Even recent text-guided methods require laborious "text-feature" mapping and fine-tuning, a process ill-suited for industrial anomaly images that often lack clear textual descriptors.

To overcome these issues, we reframe anomaly synthesis as an inverted image inpainting task. Instead of removing objects, we controllably "paint" defects onto clean backgrounds, leveraging the

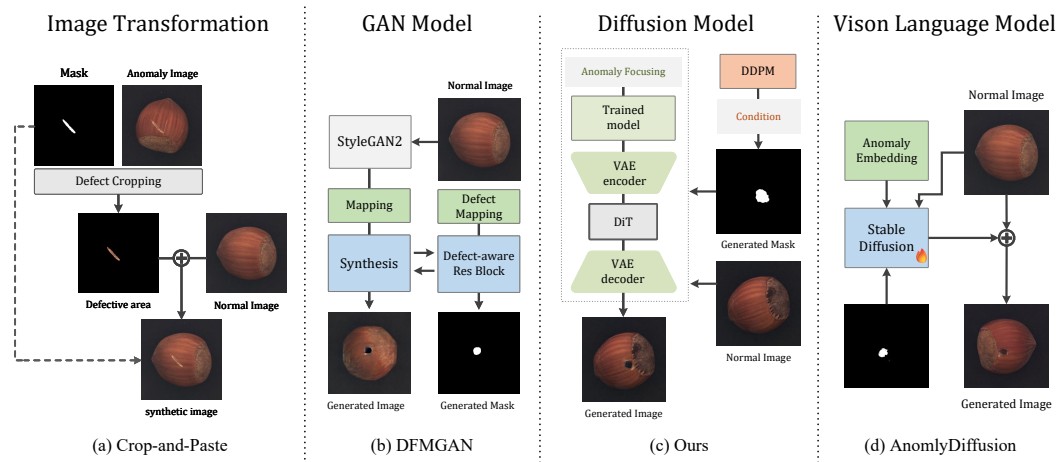

Figure 1: **Comparison of anomaly generation methods.** In contrast to existing approaches, our method provides a fully automated pipeline, generating anomalies with enhanced realism and diversity.

abundance of normal samples. Our proposed method, AnoDiT, implements this using a DiT, which excels at capturing global dependencies and generating high-fidelity textures. This approach preserves background integrity while simplifying the generation process. To ensure the realism of synthesized anomalies, we introduce two key mechanisms: a Laplacian Pyramid feature decomposition to guide the model in learning deep, multi-scale texture representations, and an Anomaly Focusing technique that uses a weighted loss on a dilated mask, compelling the model to learn both the anomaly's core features and its boundary for seamless integration.

The quality of synthetic data also depends on the annotation masks. Manual labeling is impractical, and deriving masks from generated images is complex and often inaccurate. We argue that the mask itself is a critical component for guiding synthesis and enhancing diversity. Recognizing that masks encode crucial spatial information, we propose generating them directly. We designed a conditional diffusion model that incorporates positional priors learned from the training data. This ensures that synthesized masks are not only varied in shape and size but also adhere to the plausible locations of real-world anomalies, preventing the generation of physically impossible anomalies.

In summary, this paper proposes AnoDiT, a mask-guided regional inpainting method for industrial anomaly synthesis. To our knowledge, this is the first work to adapt a non-text-guided inpainting model for this task. Our primary contributions are: (1) a DiT-based regional inpainting method that preserves background context; (2) a regional anomaly focusing mechanism with a dilated edge-weighted loss to learn anomaly features and boundaries; (3) a texture feature decomposition module to guide fine-grained texture learning; and (4) a conditional diffusion model with positional priors for generating diverse, class-conditional binary masks.

## 2 RELATED WORK

### 2.1 GENERATIVE MODELS

Generative models have evolved rapidly, with Denoising Diffusion Probabilistic Models (DDPM) (Ho et al.) largely succeeding GAN (Goodfellow et al.) due to their superior training stability and sample diversity. The introduction of the Transformer architecture in Diffusion Transformers (DiT) Peebles & Xie (2023) further advanced these capabilities, setting a new state-of-the-art in high-fidelity image synthesis.

This powerful generative prior has been widely applied to image inpainting. Context Encoder (Pathak et al., 2016) first introduced GAN to the image inpainting task, framing it as a conditional generation problem. Partial Convolution (Liu et al., 2018) proposed a conditional mask update

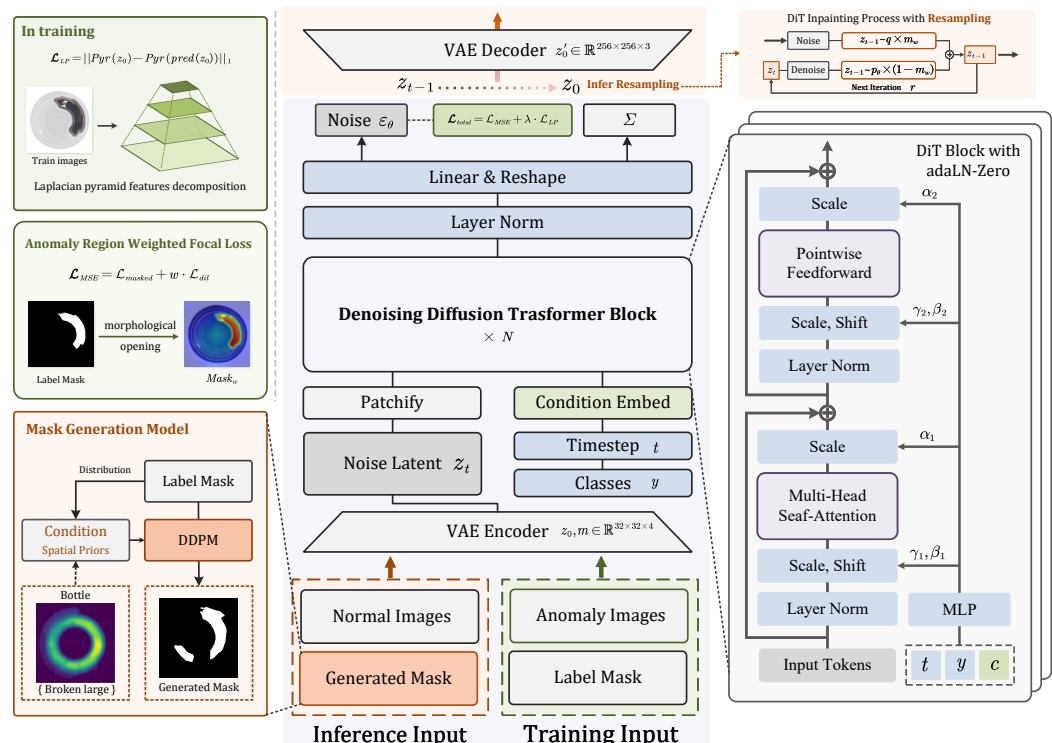

Figure 2: **The AnoDiT model architecture.** The central pipeline consists of training and inference processes. Key components include: (right) the core DiT block with adaLN-zero, (top-right) a resampling mechanism for seamless blending, and (bottom-left) a DDPM-based module generates a mask that is used to condition the inpainting process of the DiT.

mechanism, enabling the model to adaptively handle irregularly shaped missing regions. The Two-Stage Coarse-to-Fine Network (Yu et al., 2019) constructed a coarse-to-fine cascaded architecture and introduced a contextual attention mechanism to effectively model long-range dependencies. More recently, RePaint (Lugmayr et al., 2022), as the first prominent work applying diffusion models to image inpainting, unifies high fidelity and detail consistency through a conditional denoising diffusion process. As a conditional task, image inpainting has been profoundly influenced by the advancements in generative models.

## 2.2 GENRATIVE MODELS FOR INDUSTRIAL ANOMALY IMAGES SYNTHESIS

The primary objective of industrial anomaly synthesis is to overcome the challenge posed by the scarcity of authentic anomaly samples. Early methods primarily relied on traditional techniques such as Crop&Paste (Lin et al., 2021) and DefectTransfer (Lu et al., 2022). With the advancement of generative models, GAN-based anomaly generation methods gradually became the mainstream approach. Among these, An improved CycleGAN (Niu et al., 2022a) to generate defect-free templates for weakly supervised surface defect segmentation with only image-level labels. Others, like Defect-GAN (Zhang et al., 2021), employed adversarial learning for full-image or local anomaly synthesis but were often constrained by the demand for large datasets. DFMGAN (Duan et al., 2023) introduced defect-aware residual blocks into a pre-trained StyleGAN2 backbone, enabling explicit feature manipulation in few-shot settings. However, GAN-based models are difficult to train and are susceptible to mode collapse.

Recently, diffusion models have demonstrated superior realism and diversity in industrial anomaly synthesis. AnomalyDiffusion Hu et al. (2024) achieves high-quality, few-shot synthesis through latent-space and spatial anomaly embedding mechanisms, utilizing attention re-weighting to ensure alignment between the generated anomaly and its corresponding mask. AdaBLDM (Li et al., 2024a)

integrates a latent diffusion model with a trimap strategy and online decoder adaptation to effectively enhance background preservation and defect controllability. Furthermore, DualAnoDiff (Jin et al., 2025) proposed a dual-branch diffusion mechanism to generate full images and anomaly-mask pairs separately, thereby improving generation realism, diversity, and the quality of mask-anomaly alignment. Nevertheless, both GAN and diffusion-based models that generate entire images inevitably impose higher demands on the model's feature reconstruction capabilities and require a greater number of inference steps compared to local patch generation. Another limitation is the lack of explicit control over the masks, passively obtaining masks can lead to inconsistencies between the anomaly and the mask itself.

## 3 METHODS

In Fig.2, we introduce AnoDiT, a composite architecture designed for anomaly synthesis. On one hand, we leverage a probabilistic intensity prior of the mask location as a conditional control to generate diverse masks. On the other hand, guided by these generated masks, a DiT based inpainting model synthesizes a variety of anomalies onto normal images.

### 3.1 ANOMALY-FOCUSED DiT IMAGE INPAINTING

In our model, as depicted in Fig.2, an input image $z \in \mathbb{R}^{H \times W \times C}$ is first mapped to a latent representation $z_0 \in \mathbb{R}^{32 \times 32 \times 4}$ by a VAE encoder. The core of DiT is its Transformer block, which replaces the convolutional residual blocks found in U-Nets. To inject conditioning information, DiT employs an adaptive Layer Norm (adaLN) mechanism. A zero-initialized variant, adaLN-Zero, processes the conditioning embedding vector through an MLP network. While the conditioning vector is typically based on timestep $t$ and class label $y$, our model extends this by incorporating a weight-based mask condition $m_w$. This mask is input synchronously with image augmentations to guide the model toward precise segmentation of anomalous regions. The conditioning information, after being embedded and summed, is then mapped via an MLP to the affine transformation parameters, $\gamma$ and $\beta$, s well as a scaling parameter $\alpha$ for the residual connection, required for each Transformer block. Here, $\gamma$ (scale) and $\beta$ (shift) are the core components of adaLN, directly modulating the output of the Layer Norm layer. The $\alpha$ (secondary scaling) is a set of dimension-wise scaling parameters applied to the output of each sub-module just before the residual connection with its input.

During inference, our objective is to inpaint a specific anomaly, defined by a class label $y$ and a target mask $M'$, onto a defect-free image. As shown in the top-right of Fig. 2, we employ a resampling-based inpainting strategy inspired by RePaint (Lugmayr et al., 2022). The reverse sampling process initiates from pure Gaussian noise, $z_T \sim \mathcal{N}(0, I)$. For each denoising step $t \to t - 1$, we leverage the known context from the original defect-free image, $x_{norm}$. First, the latent representation of the normal image, $z_{norm} = \varepsilon(x_{norm})$, is noised to the current timestep $t$, yielding $z_{norm}, t \sim q(z_t | z_0 = z_{norm})$. Then, the latent representation for the current step, $z_t$, is formed by a composite: the known regions (outside the mask $M'$) are sourced from $z_{norm}, t$, while the unknown regions (inside the mask) are taken from the ongoing generative process's previous step, $Z_T$.

$$z'_t = (1 - M') \odot z_{norm,t} + M' \odot z_t \tag{1}$$

This blended latent representation, $z'_t$, is then denoised for a single step using the DiT model, guided by the anomaly class label y and enhanced with classifier-free guidance (CFG) (Ho, 2022). This iterative resampling enforces strong consistency between the generated anomaly and the original background, resulting in a coherent and realistic final image.

**Anomaly-Focused Loss.** Standard diffusion models offer no guarantee that the generated output will be focused on the anomalous regions. In our application, since the target inpainting areas correspond to anomalies, we design the model to focus exclusively on predicting these anomalies while disregarding other regions. To this end, and to ensure the synthesized anomalies blend seamlessly with surrounding textures, we introduce an anomaly-centric, spatially weighted loss function. This function prioritizes the anomaly itself and, secondarily, its boundary.

Given a binary ground-truth anomaly mask $M$ in the latent space, we first apply a morphological dilation operator with a kernel of size $k$ to create an expanded mask $M_{dil} = M \oplus S_k$, where $S_k$ is

the structuring element. This process forms a "boundary halo". We then construct a weight map $W$ that assigns a high weight to the original mask region $M$ and a lower weight to the halo region (the area within $M_dil$ but outside $M$). The training objective is to minimize the normalized weighted Mean Squared Error (MSE) between the predicted noise $\epsilon_\theta$ and the ground-truth noise $\epsilon$:

$$\mathcal{L}_{MSE} = \mathbb{E}_{z_0, y, \epsilon, t} \left[ \frac{1}{\sum W} \|W \odot (\epsilon - \epsilon_\theta(z_t, t, y))\|_2^2 \right] \tag{2}$$

where $\odot$ denotes the element-wise product. By minimizing this objective, the model learns to generate detailed features that are not only internally consistent but also contextually coherent at their boundaries, leading to a more realistic blend.

**Laplacian Pyramid Feature Decomposition.** Industrial anomaly textures often exhibit both low-frequency structural variations and high-frequency textural changes. To effectively model these multi-scale characteristics, we incorporate a Laplacian pyramid feature decomposition (Burt & Adelson, 1983; Denton et al., 2015) as a loss during our training phase. This enables the model to process image information across different frequency bands separately. To further suppress inconsistencies between the inpainted and visible regions, we introduce a Gaussian pyramid L1 loss in the pixel space, defined as $\mathcal{L}_{LP} = \|Pyr(z_0) - Pyr(pred(z_0))\|_1$. Here, $x_0$ is the ground-truth image, and $\bar{x}_0$ is the reconstructed image obtained by VAE-decoding the prediction from the diffusion model. The inputs are downsampled into a pyramid of $K = 4$ successive resolutions (from 256 down to 32). This loss enforces multi-scale consistency between the reconstruction and the ground truth, effectively reducing both low-frequency distortions and high-frequency artifacts.

### 3.2 POSITION-PRIOR GUIDED DIVERSE MASK GENERATION

To facilitate mask-guided anomaly synthesis, we propose a DDPM for mask generation, conditioned on a positional prior. Given that binary masks are defined solely by spatial location without complex color information, employing a sophisticated architecture like the DiT is unnecessarily complex for this task. The forward process gradually adds Gaussian noise to the data over $T$ steps, defined as $q(X_t|X_{t-1}) = \mathcal{N}(X_t; \sqrt{1 - \beta_t}X_{t-1}, \beta_t I)$, while the reverse process learns a neural network $\epsilon_\theta(X_t.t)$ to denoise the data at each step.

Many existing methods for anomaly synthesis rely on a fixed set of masks from the original dataset, without generating novel mask variations. This approach can substantially limit the diversity of the final synthesized anomalies. To address this limitation, we explicitly encode the spatial distribution of anomalies into a guidance signal. Specifically, for each anomaly class $c$, we construct a Positional prior probability map $P^c$. This prior map is derived entirely from the set of ground-truth anomaly masks $\{M_i^c\}_{i=1}^{N_c}$ available in the training data, where $N_c$ is the total number of masks for class $c$. First, we compute an empirical position frequency map, $\bar{M}^c$, by aggregating and averaging all binary masks for class $c$. This map represents the empirical frequency of anomaly occurrence at each pixel location. It is formalized as:

$$\bar{M}^c = \frac{1}{N_c} \sum_{i=1}^{N_c} M_i^c \tag{3}$$

Subsequently, this frequency map is smoothed via convolution with a Gaussian filter $G_\sigma$. This operation propagates the occurrence probabilities to neighboring regions, yielding a more robust Positional prior. The smoothed probability map, denoted as $P_{smooth}^c$, is given by $P_{smooth}^c = G_\sigma(\bar{M}^c)$, where $\sigma$ is the standard deviation of the Gaussian kernel, which controls the degree of smoothing.

## 4 EXPERIMENTS

### 4.1 EXPERIMENTAL SETUP

**Datasets.** Our experiments are conducted on the MVTec AD dataset (Bergmann et al., 2019). The MVTec AD dataset is a widely recognized benchmark in the field of industrial visual AD, comprising

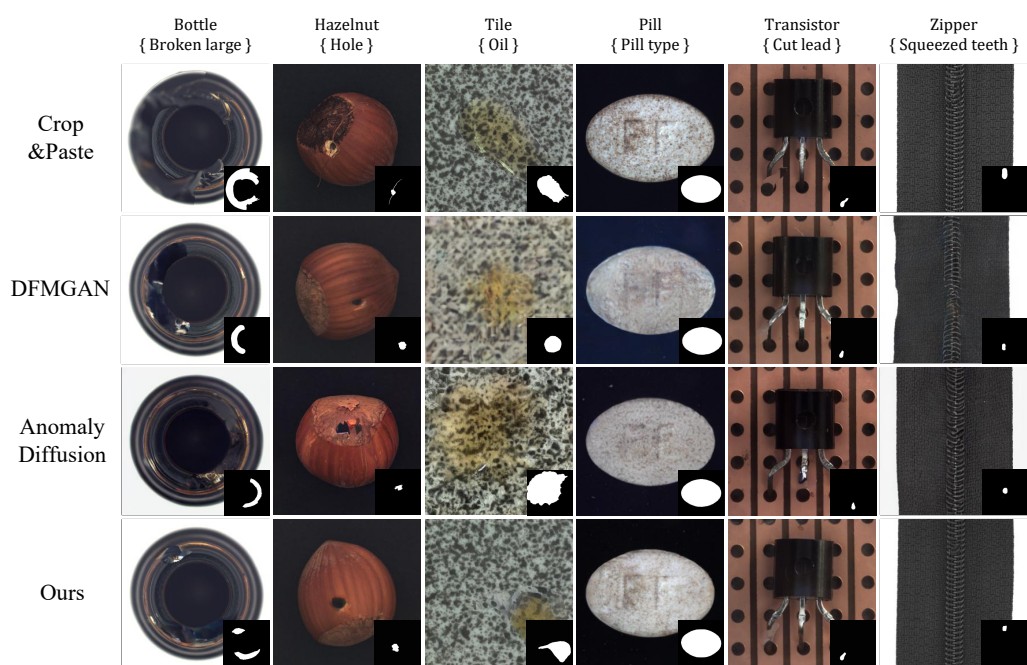

Figure 3: **Qualitative comparison of anomaly generation results on the MVTec AD dataset.** Each row showcases the generated anomalies from a different method. Columns correspond to distinct object categories, with the specific anomaly type indicated beneath each category.

5,354 images across 15 categories. On average, it contains approximately 20 anomalous images for each defect type and 200 normal images for each product class. The dataset features a variety of typical industrial anomalies, such as scratches, cracks, dents, contamination, and color deviations, and provides both image-level labels and pixel-level ground-truth masks.

**Comparison Methods.** We compare our proposed method against several recent anomaly generation techniques: Crop&Paste (Lin et al., 2021), DFMGAN (Duan et al., 2023), and AnomalyDiffusion (Hu et al., 2024). Crop&Paste generates synthetic anomalies by cropping defective regions from a limited set of samples and pasting them onto normal images. DFMGAN is a GAN specifically designed for few-shot anomaly image generation. AnomalyDiffusion utilizes a diffusion-based model and optimizes anomaly and mask embeddings via Textual Inversion.

**Evaluation Metrics.** We assess generation quality using Inception Score (IS) and Intra-Cluster LPIPS (IC-LPIPS) (Ojha et al., 2021) for perceptual similarity. For anomaly detection, we employ Area Under the Receiver Operating Characteristic Curve (AUROC), Average Precision (AP), and maximum $F_1$-Score ($F_1$-max).

**Implementation Details.** Implementation Details. We implement separate models for anomaly synthesis and mask generation using PyTorch on NVIDIA RTX 4090. For preprocessing, all images are resized to $256 \times 256$ pixels, and pixel intensities are normalized to the range $[0, 1]$ to ensure consistent input representation. All ground-truth masks are binarized for simplified processing and visualization. During inference, we generate 500 masks and corresponding anomalous images for each anomaly type. For anomaly synthesis, we employ DiT-S/2 [9] with pre-trained VAE in an image inpainting paradigm, where masks condition the defective region reconstruction. The mask generation uses class-conditional DDPM, configured with 500 timesteps and single-channel input. Both models are trained using AdamW optimizer with learning rate $1 \times 10^{-4}$.

## 4.2 Visual and Generative Metric Comparison

**Qualitative Comparison of Anomaly Generation.** The qualitative results in Fig.3 highlight the limitations of existing methods. The Crop&Paste approach is inherently constrained by the number of source anomalies in the dataset. Moreover, direct pasting can cause anomalies to extend beyond the object's boundaries. Both DFMGAN and AnomalyDiffusion are generative approaches, but GANs are notoriously difficult to train. As shown, DFMGAN fails to seamlessly integrate the generated anomaly on the 'bottle' sample. AnomalyDiffusion incorrectly fills the hole in the 'hazelnut' sample with extraneous artifacts. For smaller defects, both methods struggle to capture features accurately and sometimes fail to generate discernible anomalies.

Our method demonstrates a qualitative advantage, which we attribute to our specialized feature-capturing approach. The synthesized regions are more naturally blended with the normal image. For instance, the 'hole' feature in the 'hazelnut' class is correctly synthesized. Notably, our method can generate anomaly patterns that deviate from the training distribution, such as the two-part, disconnected mask for the 'bottle' class—a configuration not present in the original dataset but plausible in real-world settings. This shows that by generating diverse masks, our method significantly enhances the diversity of the subsequent anomaly generation model.

Table 1: **Image quality assessment for our anomaly generation method.** We report the IS and IC-LPIPS on the MVTec AD dataset.

| Category | Crop & Paste | | DFMGAN | | AnoDiffusion | | Ours | |
|---|---|---|---|---|---|---|---|---|
| | IS ↑ | IC-L ↑ | IS ↑ | IC-L ↑ | IS ↑ | IC-L ↑ | IS ↑ | IC-L ↑ |
| bottle | 1.43 | 0.04 | **1.62** | 0.12 | 1.58 | 0.19 | 1.59 | **0.20** |
| cable | 1.74 | 0.25 | 1.96 | 0.25 | **2.13** | **0.41** | 2.04 | 0.28 |
| capsule | 1.23 | 0.05 | 1.59 | 0.11 | 1.59 | **0.21** | **1.69** | 0.07 |
| carpet | 1.17 | 0.11 | 1.23 | 0.13 | 1.16 | 0.24 | **1.27** | **0.29** |
| grid | 2.00 | 0.12 | 1.97 | 0.13 | 2.04 | 0.44 | **2.30** | **0.23** |
| hazelnut | 1.74 | 0.21 | 1.93 | 0.24 | **2.13** | 0.31 | 1.83 | **0.29** |
| leather | 1.47 | 0.14 | 2.06 | 0.17 | 1.94 | **0.41** | 1.72 | 0.12 |
| metal nut | 1.56 | 0.15 | 1.49 | 0.32 | **1.96** | 0.30 | 1.40 | **0.33** |
| pill | 1.49 | 0.11 | **1.63** | 0.16 | 1.61 | **0.26** | 1.59 | 0.15 |
| screw | 1.12 | 0.16 | 1.12 | 0.14 | **1.28** | 0.30 | 1.18 | **0.39** |
| tile | 1.83 | 0.20 | 2.39 | 0.22 | **2.54** | **0.55** | 1.99 | 0.29 |
| toothbrush | 1.30 | 0.08 | 1.82 | 0.18 | 1.68 | 0.21 | **2.27** | **0.26** |
| transistor | 1.39 | 0.15 | 1.64 | 0.25 | 1.57 | 0.34 | **1.90** | **0.37** |
| wood | 1.95 | 0.23 | 2.12 | 0.35 | 2.33 | **0.37** | **2.63** | 0.23 |
| zipper | 1.23 | 0.11 | 1.29 | 0.27 | 1.39 | 0.25 | **2.02** | **0.31** |
| Average | 1.51 | 0.14 | 1.72 | 0.20 | 1.80 | 0.32 | **1.83** | 0.25 |

**Quantitative Comparison of Anomaly Generation.** In Tab.1, we report IS and IC-LPIPS scores for comparison (AnoDiffusion is AnomalyDiffuson). The results show that our method achieves superior IS scores on the majority of classes, with particularly significant improvements on the 'toothbrush' and 'zipper' categories. Although our method is outperformed on a few classes (e.g., leather, metal nut, and screw), our average IS score across all MVTec categories is higher than that of the competing methods.

**Qualitative Results for Mask Generation.** To visually assess the generated masks, we compute 2D heatmaps for each defect class of the 'Bottle' category, where pixel intensity corresponds to the anomaly probability at that location. Fig.4 displays the probability density heatmaps for both ground-truth and generated masks (100 samples), alongside their marginal distributions. The heatmaps confirm that, guided by the Positional prior, the generated masks consistently conform to the product's silhouette. In terms of diversity, the marginal distributions of our generated masks

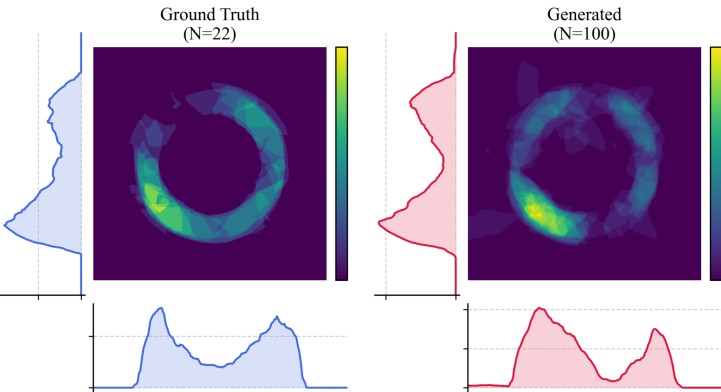

Figure 4: **Heatmap comparison of mask distributions.** This figure compares probability heatmaps of ground-truth masks (top) versus our generated masks (bottom). Projections of the marginal distributions are shown along the x and y axes for each heatmap. N indicates the number of samples. Examples from the "broken small" defect category of "Bottle".

largely align with the ground-truth distribution while still exhibiting subtle variations. This suggests our model successfully balances adherence to the object's true spatial constraints with the generation of novel and diverse mask shapes.

Table 2: **Image-level anomaly detection results on the MVTec AD dataset.** Performance is measured in terms of AUROC, AP, and $F_1$-max (higher is better).

| Category | DFMGAN | | | AnomalyDiffusion | | | Ours | | |
|---|---|---|---|---|---|---|---|---|---|
| | AUROC | AP | $F_1$-max | AUROC | AP | $F_1$-max | AUROC | AP | $F_1$-max |
| bottle | 99.3 | 99.8 | 97.7 | 99.8 | **99.9** | 98.9 | **99.9** | **99.9** | **99.1** |
| cable | 97.8 | 93.8 | 100.0 | 100.0 | 100.0 | 100.0 | 99.5 | **100.0** | 95.0 |
| capsule | 92.8 | 98.5 | 94.5 | 99.7 | 99.9 | 98.7 | **99.8** | **100.0** | **99.0** |
| carpet | 87.9 | 87.3 | 96.7 | 96.7 | **98.8** | 94.3 | **99.0** | **99.0** | 97.0 |
| grid | 90.4 | 85.4 | 98.4 | 99.5 | 98.7 | **98.7** | **100.0** | **99.0** | 95.0 |
| hazelnut | 99.9 | 100.0 | 99.0 | 100.0 | 100.0 | 100.0 | **100.0** | **100.0** | **100.0** |
| leather | 99.9 | 100.0 | **99.2** | 100.0 | 100.0 | 100.0 | **100.0** | **100.0** | 97.0 |
| metal nut | 99.8 | 99.2 | 100.0 | 100.0 | 100.0 | 100.0 | **100.0** | **100.0** | **100.0** |
| pill | 68.7 | 91.7 | 91.4 | 98.0 | 99.6 | 97.0 | **99.5** | **99.8** | **97.5** |
| screw | 64.7 | 85.3 | 96.8 | 97.9 | 95.5 | 100.0 | **99.0** | **97.0** | 95.0 |
| tile | 100.0 | 100.0 | 100.0 | 100.0 | 100.0 | 100.0 | **100.0** | **100.0** | **100.0** |
| toothbrush | 100.0 | 100.0 | 100.0 | 100.0 | 100.0 | 100.0 | **100.0** | **100.0** | **100.0** |
| transistor | 92.5 | 88.9 | 100.0 | 100.0 | 100.0 | 100.0 | **100.0** | **100.0** | **100.0** |
| wood | 99.4 | 98.8 | 98.4 | 99.4 | 99.4 | **98.8** | **100.0** | **100.0** | 98.0 |
| zipper | **99.9** | 99.4 | **99.9** | **99.9** | **99.9** | 99.4 | 100.0 | 100.0 | 97.0 |
| Average | 93.5 | 95.2 | 97.5 | 99.4 | 99.5 | **98.5** | **99.8** | **99.6** | 98.0 |

## 4.3 ANOMALY DETECTION PERFORMANCE

To quantitatively evaluate the effectiveness of the generated images for improving downstream task performance, we conducted a comparison for image-level AD on the MVTec AD dataset. Tab.2 summarizes the performance of each method, evaluated using standard metrics in the field: AUROC, AP, and $F_1$-max. For this experiment, we generated 1,000 synthetic anomaly images for each anomaly class to train a unified UNet (Ronneberger et al., 2015) segmentation network. The re-

sults in the table show that our method achieves competitive or superior performance across the vast majority of categories.

Table 3: **Ablation study on the contribution of each model component.** The full model achieves the best performance across all metrics.

| Components | | Performance Metrics | | | | |
|---|---|---|---|---|---|---|
| AF | LP | IS $\uparrow$ | IC-LPIPS $\uparrow$ | AUROC $\uparrow$ | AP $\uparrow$ | $F_1$-**max** $\uparrow$ |
| $\times$ | $\times$ | 1.19 | 0.18 | 90.6 | 89.2 | 87.0 |
| $\checkmark$ | $\times$ | 1.39 | 0.11 | 95.9 | 96.0 | 94.9 |
| $\times$ | $\checkmark$ | 1.53 | 0.20 | 95.4 | 94.6 | 96.7 |
| $\checkmark$ | $\checkmark$ | **1.83** | **0.25** | **99.8** | **99.6** | **98.0** |

## 4.4 ABLATION STUDY

We performed an ablation study to assess the effectiveness of the anomaly-focusing and feature-decomposition mechanisms in our proposed model. We individually removed the anomaly-focusing weighted loss function and the Laplacian feature decomposition mechanism, reverting to a standard MSE loss for each case. Apart from the loss function, both configurations utilized the identical model architecture, dataset, and training hyperparameters.

Tab.3 illustrates the contribution of the Anomaly Focusing (AF) and Laplacian feature decomposition (LP) components. The results clearly demonstrate that removing either component leads to a significant degradation in performance. Specifically, using the AF module alone primarily enhances generation quality, whereas the LP module alone provides a greater advantage on classification-related metrics. Optimal performance is achieved by combining both modules.

## 5 CONCLUSION

In conclusion, we introduced AnoDiT, a mask-guided, anomaly-centric inpainting model based on the DiT. This model recalibrates the learning objective of the diffusion process to focus exclusively on anomalous regions. It preserves the background context by integrating it from the forward diffusion into the reverse diffusion process and leverages a Laplacian feature decomposition to learn anomaly characteristics, thereby achieving superior performance in targeted anomaly restoration. Furthermore, to obtain a more diverse set of masks, we developed a DDPM-based mask generation model guided by a positional prior. This allows for adjustable guidance strength based on product characteristics, ensuring that the generated anomaly locations adhere to realistic spatial constraints. By incorporating a larger and more varied set of masks into the synthesis process, we significantly enhance both the quantity and diversity of our synthetic data.

A key challenge for generative models trained on limited datasets is their tendency to replicate the original data distribution, leading to a lack of diversity in synthesized features. While our mask-generation approach enhances diversity, it also underscores the importance of the quantity and quality of ground-truth masks in the training dataset. As data-driven machine learning continues to gain prominence, our proposed framework serves as a generalizable paradigm. Beyond its application in industrial inspection, it can be extended to other domains rich in normal samples but scarce in anomalies, such as medical imaging, remote sensing, and materials science, following domain-specific training.

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
