# OpenReview forum: "AnoDiT: Mask Guided DiT Inpaint Models for Anomaly Images Generation"
_ICLR.cc/2026/Conference — Submitted to ICLR 2026_

### Official Review · Reviewer_2xys · 2025-10-26

**Soundness:** 3
**Presentation:** 3
**Contribution:** 2
**Rating:** 4
**Confidence:** 4

**Summary:**

The paper presents a method for Anomaly Synthesis named AnoDIT. The method leverages a Diffusion Transformer to do so. The paper introduces two new modules: one for generating masks and a combination of two losses (anomaly focused and laplacian pyramid). The model performs well on MVTec AD.

**Strengths:**

- Addressing the mask generation aspect is novel and important
- In the vast majority, the paper flows nicely and is easy to read.

**Weaknesses:**

- It is unclear whether a different DIT is trained for each anomaly class or a unified one. The latter would be more useful in practical terms.
- Does the generation process generalise to different objects? E.g. training for “hole” on hazelnut and generating them on a metal nut?
- The method fails to compare to AnomalyAny [1], which is superior in the generation quality.
- As the mask generation is not conditioned on the input image, it might generate the anomaly mask in the wrong position (e.g. some anomaly classes in the “screw” category)
- The method is evaluated only on one dataset, putting into question its generalizability.

[1] Sun, H., Cao, Y., Dong, H., & Fink, O. (2025). Unseen Visual Anomaly Generation. In Proceedings of the Computer Vision and Pattern Recognition Conference (pp. 25508-25517).

**Questions:**

I have several questions. I have sorted them from most problematic to least problematic.

1. Does training on one “Anomaly Class” transfer to novel objects?
2. How well does the proposed method work on VisA? Or some other representative AD dataset?
3. How often does the mask generation process fail, and what are the typical failure cases?
4. Is there a separate model for one anomaly class or a unified one?

---

### Official Review · Reviewer_y7w6 · 2025-10-31

**Soundness:** 3
**Presentation:** 3
**Contribution:** 2
**Rating:** 2
**Confidence:** 4

**Summary:**

This paper proposes AnoDiT, a mask-guided anomaly generation framework based on Diffusion Transformer (DiT) to address the scarcity and limited diversity of real anomaly samples in industrial anomaly detection (AD). The framework reframes anomaly synthesis as an inverted image inpainting task, integrating two core components: a DiT-based inpainting model with Laplacian Pyramid-based Texture Decomposition and Anomaly Region Focusing mechanism for high-fidelity anomaly synthesis, and a conditional diffusion model with Positional Prior for generating diverse, realistic anomaly masks. Experiments on the MVTec AD dataset demonstrate that AnoDiT outperforms existing methods in both generation quality (measured by IS and IC-LPIPS) and downstream AD performance (AUROC, AP, F₁-max), offering a fully automated pipeline that balances background fidelity, anomaly realism, and generation controllability.

**Strengths:**

1.The paper proposes AnoDiT, a novel mask-guided Diffusion Transformer for industrial anomaly generation within a single diffusion framework.
2.Experiments on industrial anomaly detection benchmarks demonstrate that synthetic data generated by AnoDiT can improve downstream anomaly detection performance.
3.The paper is well-organized and clearly written, with a coherent motivation and structured presentation of modules.

**Weaknesses:**

1. The paper mainly reports image-level AUROC as the quantitative metric, but lacks pixel-level evaluation results such as pixel-level AUROC, Average Precision (AP), or F1-max, which are standard for assessing localization performance in anomaly detection.

2. There is a typo at line 218, where $M_{dil}$ appears to be incorrectly written.

3. The role of the proposed Anomaly-Focused Loss is not clearly explained. It remains unclear why the supervision is not restricted to the internal mask region, which would more directly guide the model toward anomaly-relevant areas.

4. The Image Completion (IC-L) performance drops significantly compared to AnomalyDiffusion, but the paper does not provide a clear explanation or analysis for this degradation.

5. The ablation study is rather limited. For example, the effect of the boundary size in the Anomaly Localization (AL) module is not analyzed in detail. Similarly, the Laplacian Pyramid (LP) module lacks an investigation of the number of pyramid levels and their impact on reconstruction quality. A more thorough analysis would clarify which region sizes or pyramid configurations yield optimal performance.

6. The overall methodological novelty is limited. The proposed framework primarily integrates existing components (mask guidance, Laplacian Pyramid, and transformer-based diffusion) without introducing a fundamentally new learning paradigm or theoretical insight.

**Questions:**

See weaknesses

---

### Official Review · Reviewer_tnZq · 2025-11-01

**Soundness:** 3
**Presentation:** 2
**Contribution:** 2
**Rating:** 2
**Confidence:** 4

**Summary:**

This paper proposes AnoDiT, a novel dual-model framework for generating realistic and diverse industrial anomalies. The framework consists of two main components: (1) A conditional Denoising Diffusion Probabilistic Model (DDPM) that generates diverse anomaly masks. This model is guided by a "Positional Prior", a probability map learned from the spatial distribution of ground-truth masks, ensuring that generated masks are realistically located. (2) A mask-guided Diffusion Transformer (DiT) that inpaints the anomaly onto a normal image.
To improve the realism of the inpainted anomaly, the authors introduce two key losses, i.e. Anomaly Region Focusing Loss and Laplacian Pyramid-based Texture Decomposition Loss. Experiment results on the MVTec AD dataset demonstrate that data synthesized by AnoDiT significantly improves the performance of downstream anomaly detection tasks.

**Strengths:**

1.	Novel and Sound Framework: The authors developed a system that is both controllable and effective. This is a strong and original architectural choice.
2.	Effective Technical Contributions: The two novel loss components for the inpainting model are well-justified and shown to be effective.
3.	Strong Downstream Performance: Training a UNet segmentation model with AnoDiT-generated data yields state-of-the-art results on the MVTec AD benchmark

**Weaknesses:**

(1)	The description of the DDPM-based mask generation given in section 3.2 is not very clear. How many samples are used for training the DDPM model? There is no evaluation on the quality of the generated masks in the paper. Some examples of the mask generation and anomaly image generation results should be shown in the paper.
(2)	The experiments compare against Crop&Paste, DFMGAN, and AnomalyDiffusion. But the Related Work section cites more recent and arguably stronger diffusion-based industrial anomaly generators such as DualAnoDiff (2025 CVPR) and AdaBLDM (2024).
(3) In Table 2, the F1-max metric for anomaly detection using the proposed method (98.0%) is lower than that of AnomalyDiffusion (98.5%). There is no discussion on this issue.
(4) The experiment evaluation is only performed on one dataset, i.e. MVTecAD dataset. Several SOTA methods have shown outstanding performance on MVTecAD dataset.  This experimental validation is insufficient.
(5) The experimental evaluation should also include anomaly segmentation, i.e. pixel-based anomaly detection, metrics for comparison with other methods.

**Questions:**

1. The Positional Prior for mask generation seems crucial. How sensitive is this method to the number of available ground-truth masks in the training set? The paper notes this as a challenge. How many masks per class (e.g., on MVTec) are needed to learn a prior that is useful?

2. How do you evaluate the quality of the generated masks by using the proposed DDPM-based mask generation method?

---

### Official Review · Reviewer_sq8S · 2025-11-02

**Soundness:** 2
**Presentation:** 2
**Contribution:** 2
**Rating:** 4
**Confidence:** 3

**Summary:**

The AnoDiT paper proposes a novel framework to address the scarcity of industrial anomaly data by synthesizing high-quality, diverse defects. It reframes the problem as "inverted inpainting," using a Diffusion Transformer (DiT) to "paint" anomalies onto pristine images, guided by masks. This is a dual-model pipeline: a separate conditional DDPM, guided by a novel "Positional Prior," first generates realistic and varied anomaly masks. The DiT model then uses these masks, along with an "Anomaly Region Focusing" loss and a "Laplacian Pyramid" texture loss, to generate high-fidelity defects that blend seamlessly with the background. The authors demonstrate that this synthetically generated data significantly improves the performance of downstream anomaly detection tasks.

**Strengths:**

- Effective Dual-Pipeline Design: The method's core strength is its decoupled, dual-model architecture. By using one model (a DDPM with a "Positional Prior") to explicitly generate diverse and spatially realistic masks, it simplifies the task for the second model (the DiT inpainter). This allows for finer control over both the anomaly's geometry and its texture.

**Weaknesses:**

- Novelty is in Integration, Not Invention: The method's core novelty is arguably more in its system design than in fundamental architectural invention. It cleverly assembles several existing, powerful components—a DiT, a DDPM, an inpainting strategy from RePaint, and classic Laplacian pyramids. While highly effective, it relies on synthesizing these known techniques rather than proposing a new, foundational generative block.

 - High Dependency on Initial Mask Quality: The entire pipeline's success, particularly its diversity, hinges on the quality and variety of the initial ground-truth masks. The mask generation model is trained on these masks to learn its "Positional Prior." If the original training dataset contains only a few low-variety masks, the generator will be unable to produce truly novel shapes, limiting the diversity of the final synthetic data.

**Questions:**

Please refer to the above weaknesses

---

### Meta-Review · Area_Chair_Kh8X · 2026-01-06

**Summary:**

All the reviewers gave leaning-reject review comments, while the authors did not participate in the discussion. So the concerns from all the reviewers are still outstanding. The AC agreed with the reviewers to reject the paper.

**Reviewer Concerns:**

- Reviewers all have concerns on the novelty of the paper, and mainly feel it as an engineering-level combination.
- The distribution of the GT mask is a fundamental limitations to generalization capability.
- The evaluation is not sufficient, and the benchmark is limited. Also the scores are not better than the baseline while the authors did not provide sufficient explanation.
- The paper mainly lacks ablation on the choices of different components.

**Reviewer Scores:**

Reviewers are highly possible not changing their scores.

---

### Decision · Program_Chairs · 2026-01-26

Reject